# *Bemisia tabaci* (Hemiptera: Aleyrodidae): What Relationships with and Morpho-Physiological Effects on the Plants It Develops on?

**DOI:** 10.3390/insects13040351

**Published:** 2022-04-01

**Authors:** Alessia Farina, Antonio C. Barbera, Giovanni Leonardi, Giuseppe E. Massimino Cocuzza, Pompeo Suma, Carmelo Rapisarda

**Affiliations:** 1Applied Entomology Section, Department of Agriculture, Food and Environment (Di3A), University of Catania, 95123 Catania, Italy; cocuzza@unict.it (G.E.M.C.); suma@unict.it (P.S.); carmelo.rapisarda@unict.it (C.R.); 2Agronomy and Field Crops Section, Department of Agriculture, Food and Environment (Di3A), University of Catania, 95123 Catania, Italy; antonio.barbera@unict.it (A.C.B.); giovanni.leonardi@outlook.com (G.L.)

**Keywords:** whitefly, vegetable crops, plant morphology, plant physiology, trophic interactions

## Abstract

**Simple Summary:**

*Bemisia tabaci* (Gennadius) (Hemiptera: Aleyrodidae) has a cosmopolitan distribution, and it is a feared pest of many agricultural crops. It is a complex of numerous genetically differentiated species, most of which may rapidly acquire insecticide resistance, consequently making their control problematic. This study aims to improve knowledge on the direct damage of this pest, as well as its impact on the main traits of vegetable crops. Overall, the results confirm how different host plants display variable susceptibility to *B. tabaci* infestation and explain trophic links between plant and pest forecasting plant growth and development under *B. tabaci* presence.

**Abstract:**

Although many crops have developed several adaptation mechanisms that allow them to defend against limiting factors, some biotic and abiotic stresses may cause reversible or irreversible changes in plants. Among the biotic stresses, the whitefly *Bemisia tabaci* (Gennadius) (Hemiptera: Aleyrodidae) is probably one of the main important pests that negatively affect several vegetable crops that are grown in greenhouses. The present study evaluated its impact on the morphology and physiology of two solanaceous plants, i.e., tomato (*Solanum lycopersicum* L.) and eggplant (*S. melongena* L.), under laboratory conditions. The results showed that, for tomatoes, plant height, shoot dry weight, leaf area, and indirect chlorophyll content were strongly reduced in infested plants, compared to the uninfested control, by 39.36%, 32.37%, 61.01%, and 37.85%, respectively. The same has been shown for eggplant, although the reduction percentages of plant height, root dry weight, and indirect chlorophyll content were less marked (i.e., 16.15%, 31.65%, and 11.39%, respectively). These results could represent interesting information for a better understanding of the *B. tabaci* influence on plant growth, as well as for the development of management strategies to successfully control its infestations in a cropping system.

## 1. Introduction

The economic importance of the plant family *Solanaceae* has been extensively discussed [1], as well as the role these plants have had in the progress of traditional cultures and civilizations [2]. Among them, tomato (*Solanum lycopersicum* L.) and eggplant (*S. melongena* L.) are very common vegetable crops, widely spread worldwide and known for their culinary, medicinal, and ornamental uses [2,3].

Plants interact with the environment, and any unfavorable conditions may impose stress and reduce their growth and development [4]. Although plants have developed several adaptation mechanisms that allow them to defend against limiting factors, biotic and abiotic stresses may generate reversible or irreversible changes in their morphology and physiology; in cultivated plants, this may lead to losses in crop production and yield [5].

The whitefly *Bemisia tabaci* (Gennadius) (*Hemiptera*, *Aleyrodidae*) is a species complex with a worldwide distribution, considered a serious pest of many agricultural crops [6]. Throughout its life cycle, it feeds on the leaves, underside of the host plants, causing direct and indirect damages by piercing leaves, sucking sap, and producing honeydew (on which sooty mould develop), as well as altering the growth, photosynthesis, and chemical and phenological processes [7,8], in addition to transmitting various plant viruses (more than 350) that cause serious diseases [9]. Indeed, *B. tabaci* consists of numerous genetically and biologically different cryptic species, more or less invasive, frequently with a different impact on many economically important crops [6,10].

Control of *B. tabaci* is problematic because of the numerous generations it annually develops and its ability to rapidly acquire resistance to insecticides [11]. Among various applicable control strategies, the use of tolerant varieties of the host plants is one of the cornerstones for the management of this pest [12]. Many studies on the food preferences of *B. tabaci* and host suitability among different plant varieties have been carried out over the last years [5,12,13,14,15]. Other research focused on the effect of plant development (e.g., plant age and size) on whitefly infestation and reproductive activity [16]. However, the combination of physiological and morphological changes, induced by *B. tabaci*, to the infested host plants is still poorly investigated. In the present study, the impact of the *B. tabaci* (MED) species, Q2 subclade, on some of the less investigated traits (e.g., indirect chlorophyll content, root dry weight, etc.) of whole eggplant and tomato plants was evaluated, in order to provide basic knowledge in the frame of research aimed at forecasting the growth and development of these plants under pest pressure.

## 2. Materials and Methods

The study was carried out at the laboratories of the Applied Entomology section of Di3A (Department of Agriculture, Food and Environment), University of Catania, in the period December 2020–June 2021.

To assess the impact of *B. tabaci* on the host plants growth, fourteen young tomato plants (*Solanum lycopersicum* L. cv. Dovizio) and fourteen eggplant plants (*S. melongena* L. cv. Gloria), with six fully expanded leaves, were used in the test. Experimental plants were grown from seeds germinated and raised in polystyrene planting tray in the nursery. Then, the seedlings were individually transplanted into black plastic pots (10 cm × 10 cm × 12 cm), using a professional potting soil specific for vegetable sowing, and maintained under controlled environment at the laboratory (T = 25 ± 2 °C; R.H. 65 ± 5% and photoperiod of 10L:14D h) throughout the experiment. Each potted plant was then confined in a netted cage (Length × Width × Height: 25 cm× 25 cm× 70 cm), representing a replication. Four weeks after transplanting, seven of each plant species were artificially infested, collecting ten pairs of newly emerged whitefly adults (<24 h old) from the insectarium and releasing them on the floor, as well as in the center of each cage. The whitefly adults were allowed to lay eggs for five days, before being removed from the cages by a mouth aspirator (John W. Hock Company, Gainesville, FL, United States); to ensure that oviposition had occurred, the number of eggs laid was counted on three leaves/plant, using a stereomicroscope (Olympus Optical Co., Ltd., Tokyo, JP, Japan, SZX-ILLK200). All plants were watered twice a week.

The *Bemisia tabaci* adults used in the experiment were originally collected in September 2020 from an eggplant crop grown under greenhouse in south-eastern Sicily (province of Ragusa, 36.97134 lat.; 14.424505 long). The specimens were maintained on eggplant plants, reared in laboratory, under controlled environmental conditions (25 ± 2 °C, RH 65 ± 5%, and a photoperiod of 14L:10D h). Before running the test, the species identity of *B. tabaci* has been genetically attained on about 30 whiteflies, collected from the rearing described above. To this aim, the total DNA was extracted from single individuals, following the method described by Walsh [17] and De Barro and Driver [18]. The mitochondrial cytochrome oxidase I (mt COI) gene (about 710 bp) was amplified using universal primers LCO1490 and HCO2198 [19,20,21]. For each sample, the 10 μL reaction volume contained 5 μL of FailSafe™ 2X PreMixes buffers (Lucigen, Middleton, WI, United States), 3.75 μL of DNA, 0.25 μL of taq polymerase, and 0.5 μL of each forward and reverse primer. The PCR was performed with initial denaturation at 96 °C for 5 min, followed by 35 cycles, each consisting of denaturation for 45 s at 96 °C, annealing for 60 s at 45 °C, with final extension for one minute at 72 °C, followed by final extension for 10 min, at 72 °C. PCR-amplified products (10 μL) were visualized with 0.9% agar-gel electrophoresis (5 μL), and products with the target fragment were selected for sequencing. Successfully amplified DNA (5 μL) was purified and sequenced by BMR genomics. Identity of *B. tabaci* MED were based on more than 99% of the sequence similarity, obtained by NCBI blast comparison.

To assess the effects of *B. tabaci* on *S. lycopersicum* and *S. melongena* growth, the height of the plants, indirect chlorophyll content, fresh and dry plant biomass (roots and shoots), and leaf area were measured at the end of the experiment, when whitefly adults of the first generation were detected inside the cages (i.e., after about 28 days form the insect’s release). Plant height, expressed in centimeters, was measured with a ruler; the fresh weight of the plant was expressed in grams, cutting and weighting shoots and roots with a high precision balance (ORMA BC 1000, Orma srl, Milan, IT, Italy; resolution 0.1 g). Regarding dry weights, the biomass was oven-dried (Thermo Fisher Scientific™, Langenselbold, DE, Germany Heratherm OGS100) at 65 °C, until a constant weight was reached in three days and, finally, weighed and expressed in grams, as well. The indirect chlorophyll content measurements were performed using a Soil Plant Analysis Development (SPAD-502, Minolta, Sakai (Osaka), Japan) chlorophyll meter on three leaves per plant, which were at the principal growth stage 1 leaf development, according to the BBCH scale [22]. The obtained values, expressed in SPAD units, proportionally reflect the amount of chlorophyll present in the leaf [23]. The plant leaf area, expressed in cm^2^, was determined by ImageJ software (Wayne Rasband—Services Branch, National Institute of Mental Health, Bethesda, MD, United States), which processed the photos shot by a digital camera (48-megapixel).

### Data Analysis

The impact of the whitefly on the host plants development was expressed as the percentage reduction of the values of the parameters considered, which were calculated as follows:$$\%\;\mathrm{decrease}=\frac{\mathrm{Uninfested}\;\mathrm{plants}\;\mathrm{value}\;-\;\mathrm{Infested}\;\mathrm{plants}\;\mathrm{value}}{\mathrm{Uninfested}\;\mathrm{plants}\;\mathrm{value}}\;\times \;100$$

The data, related to the different plant’s parameters selected, were subjected to analysis of variance (ANOVA), and mean comparisons were performed according to the Fisher’s LSD test. Statistics were carried out by using the program Statistica (StatSoft, TIBCO Software Inc., Tulsa, OK, USA).

## 3. Results

Molecular analysis identified the species used in the experiment as *B. tabaci* MED, Q2 subclade, confirming the results obtained by Parrella [24], who asserted this as the most widespread species living on solanaceous in the Mediterranean area.

After five days from the release of the adults, the mean number of eggs laid on the lower surface of each of the three selected leaves were 46.11 ± 10.08 (average: 2.2 eggs/cm^2^) and 51.33 ± 4.45 (average: 1.8 eggs/cm^2^) for tomato and eggplant, respectively, confirming that an equivalent oviposition level occurred in both the host plant species (Figure 1).

For tomatoes, the heights of the plants were significantly higher in the control ones (*F*_6_ = 27.40; *p* < 0.01), with averages of 61.51 ± 9.80 and 37.30 ± 7.03 cm in the uninfested and infested ones, respectively (Figure 2a), with a reduction of 39.36%. Comparing the shoots dry weight of infested and uninfested plants, the data show a statistically different effect, too (*F*_6_ = 7.58; *p* < 0.05) (Figure 2b), with a significant biomass reduction, in the presence of *B. tabaci*, equal to 32.37%. Similarly, the presence of the whitefly caused a reduction in the plant leaf area of 61.01%, with mean values that are statistically different, when infested and non-infested plants are compared (*F*_6_ = 8.10; *p* < 0.05) (Figure 3a). As expected, the SPAD values followed the same trend: in fact, the indirect chlorophyll content was significantly influenced (*F*_6_ = 61.48; *p* < 0.01) by whitefly infestation that, overall, caused a 37.85% reduction (Figure 3b). By contrast, the root dry weight was not significantly affected by the presence of the whitefly (Figure 3c).

For eggplant, the results showed that plant height, root dry weight, and indirect chlorophyll content were significantly different between the treatments. In infested plants, the height (28.53 ± 2.22 cm) and root dry weight (0.08 ± 0.02 g) were significantly lower (*F*_6_ = 36; *p* < 0.01 and *F*_6_ = 30.24; *p* < 0.01) than in uninfested plants, with reduction percentages of 16.55% and 31.65%, respectively (Figure 4a and Figure 5c). Similarly, the indirect chlorophyll content values followed the same trend; in fact, the whitefly infestation significantly reduced this parameter (*F*_6_ = 14.66; *p* < 0.01) by 11.39%, with an average of 30.61 ± 3.67 SPAD units in the infested plants (Figure 5b). By contrast, the leaf area data and shoot dry weight were not significantly affected by the presence of the whitefly (Figure 4b and Figure 5a).

## 4. Discussion

It is well-known that leaves represent the major organ for solar radiation interception and photosynthetic sources in plants; in fact, plant growth and physiological processes controlling yield and dry matter production highly depend on their health and activity.

*B. tabaci* is one of the most harmful insect pests, due to its direct and indirect injures to plants, which affect the yield and quality of products [7,8,25].

The present research shows how *B. tabaci* MED infestation has a significant impact on those parameters less investigated, so far (e.g., indirect chlorophyll content, root dry weight, etc.), of whole eggplant and tomato plants. In fact, previous works, dealing with *B. tabaci* MED, only consider physiological (e.g., gas exchange and chlorophyll fluorescence) and biochemical (enzymes, phenols, and flavonoids, only on leaves) aspects, but not the combined effects of whitefly infestation on plant morphology [26,27]. In regard to the height of the two solanaceous plants, our results are in line with the findings by Islam [28], who reported that some plant-growth parameters of three eggplant varieties were negatively affected by *B. tabaci*, with a maximum reduction percentage of plant height equal to 20.6% in the “Dafeng” variety. According to Li [29], infestation by *B. tabaci* in the Middle East–Asia Minor 1 (MEAM1) species significantly inhibited the growth of tobacco plants, with plant height reductions of 28.5% and 32.7%, noted at 14 and 20 days after the start of infestation, respectively, compared to the uninfested control plants.

In particular, in our test, the shoot dry weight in tomato plants was reduced by the presence of the whitefly. Comparable findings were observed in tobacco leaves whose dry weight was significantly reduced after infestation by *B. tabaci* MEAM1 [29].

Moreover, we recorded how the indirect chlorophyll content values were greater in uninfested tomato and eggplant plants, when compared with the infested ones. Buntin [27] described that feeding activity, by both *B. tabaci* nymphs and adults, reduces leaf chlorophyll content and negatively affects the rates of leaf transpiration and photosynthesis in tomato plants. Sap suction, by the whitefly, induces chlorosis and increases stomatal resistance, which is negatively associated with photosynthetic rates, because leaf stomatal resistance indicates closure of stomata and limited gas exchange [29]. Contextually, the eggs being laid on the lower surface of the leaves by the whitefly females significantly decrease the stomatal conductance, because they cover the stomata and block their access to light and carbon dioxide [30]. Similar results were also observed in other studies, in which *B. tabaci* infestation reduced cotton foliar photosynthetic rates [31,32,33].

In our experiment, the leaf area in tomato plants was statistically different between the infested and uninfested control. As to this aspect, Chand [30] reported how eggs deposition by *Aleurodicus dispersus* Russell (Hemiptera: Aleyrodidae) on cassava (*Manihot esculenta* Crantz) reduces the effective area of leaves, lowering the overall productivity. By contrast with what has been reported on eggplant by Islam [34], who states that the whitefly presence lowered the effective leaf area (although no mention is provided on the density of whitefly), in our study, the leaf area and shoot dry weight of eggplant were not affected by the presence of the insect. This can be explained by assuming that, in our experiment, the lack of significant effects on both these parameters could be related to the low density of the whitefly. Indeed, the density of 2.2 eggs/cm^2^, recorded on the smaller leaves of tomatoes, was enough to negatively affect its leaf area and shoot dry weight, compared to what we noted on eggplant, where an average density of 1.8 eggs/cm^2^ did not affect these parameters. It is, in fact, well-known that leaf size and shape can vary from plant-to-plant, as well as among varieties, and this may differently affect the tolerance to whitefly infestation among different host plant species [5,12,35,36,37].

## 5. Conclusions

The present research points out that one single generation of *B. tabaci* moderate infestation can lead to an important impact on both the morphology and physiology of tomato and eggplant plants. The overall results from this work are useful to describe the trophic relations between plants and *B. tabaci* MED. However, further investigations are needed to analyze the preferences and behavior of the whitefly on host plants, as well as identifying the possible role played by volatile organic compounds and nutritional quality of the selected plant varieties in influencing the activity of this insect.

## Figures and Tables

**Figure 1 insects-13-00351-f001:**
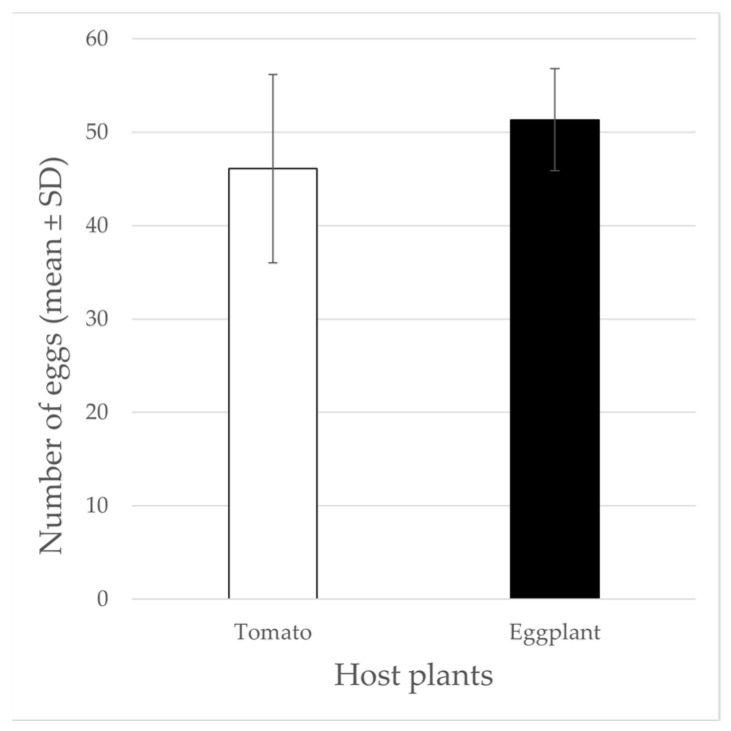
Mean number of eggs laid on the lower surface of each of the three selected leaves per plant.

**Figure 2 insects-13-00351-f002:**
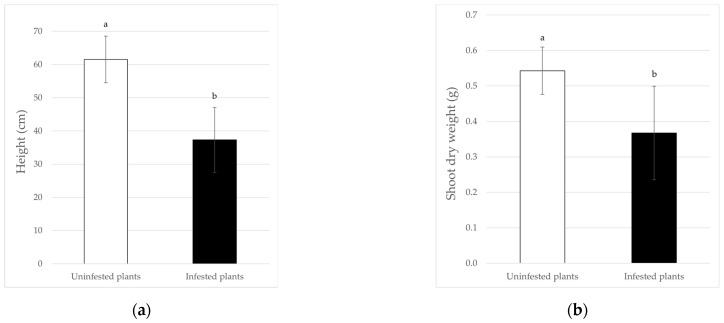
Incidence of *B. tabaci* infestation on the height (**a**) and shoot dry weight (**b**) of tomato plants. Different letters indicate statistically significant differences at *p* < 0.01 (**a**) and *p* < 0.05 (**b**).

**Figure 3 insects-13-00351-f003:**
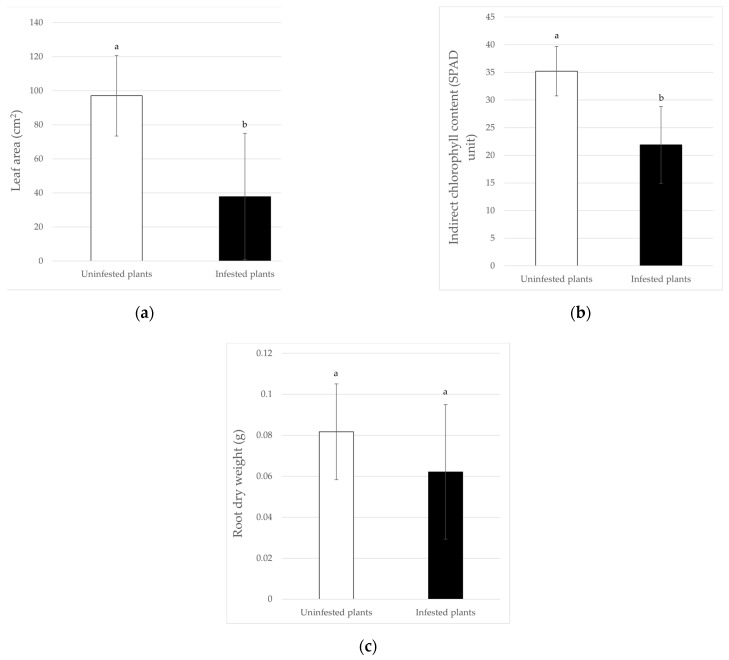
Effects of *B. tabaci* infestation on the leaf area (**a**), indirect chlorophyll content (**b**), and root dry weight (**c**) of tomato plants. Different letters indicate statistically significant differences at *p* < 0.05 (**a**) and *p* < 0.01 (**b**).

**Figure 4 insects-13-00351-f004:**
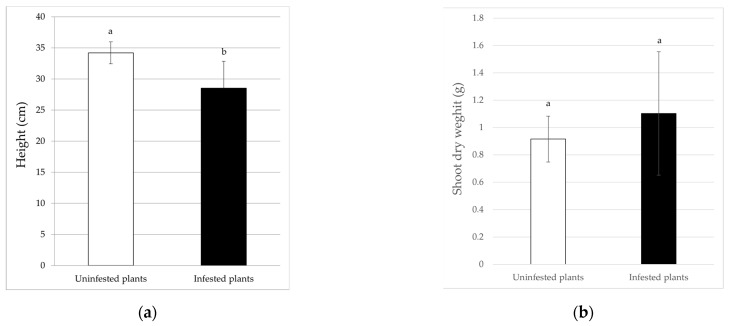
Incidence of *B. tabaci* infestation on the height (**a**) and shoot dry weight (**b**) of eggplant plants. Different letters indicate statistically significant differences at *p* < 0.01.

**Figure 5 insects-13-00351-f005:**
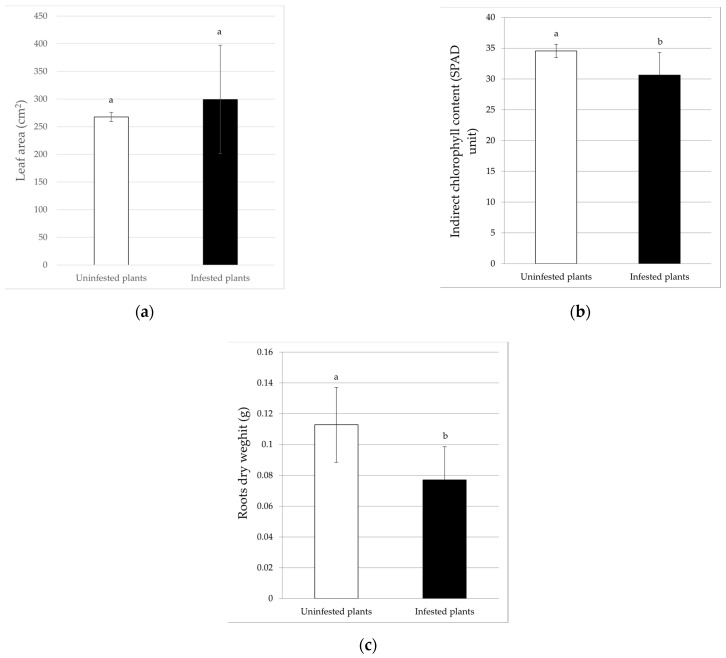
Effects of *B. tabaci* infestation on the leaf area (**a**), indirect chlorophyll content (**b**), and root dry weight (**c**) of eggplant plants. Different letters indicate statistical differences at *p* < 0.01.

## Data Availability

The data presented in this study are available on request from the corresponding author.

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
