# Peer review of "Bemisia tabaci (Hemiptera: Aleyrodidae): What Relationships with and Morpho-Physiological Effects on the Plants It Develops on?"

_insects, 2022, doi:10.3390/insects13040351_

Round 1

Reviewer 1 Report

It seems that the experiments in this work have been conducted correctly; however, I find the following problems with the present form of the manuscript:

1) The novelty of the results remains doubtful. It is not the first study about the effects of Bemisia tabaci on either tomato or eggplant species; therefore, it would be useful to review the already known data about B. tabaci effects on both plant species (tomato and eggplant) in Introduction. Especially, it should be stressed in which aspects this work is different from the previous works on the same topic.

2) The arrangement of graphs in the Results, I think, should be changed. In the present version, for tomato, height is grouped with shoot dry weight (Fig. 2) and leaf area is grouped with chlorophyll content (Fig. 3); however, for eggplant, height is grouped with root dry weight and chlorophyll content (Fig. 4), and leaf area is grouped with shoot dry weight (Fig. 5). I suggest that uniformity between the results from two plant species would be helpful. Besides, I did not manage to find what happened to root dry weight in tomato.

3) At the end of Discussion, the authors make a valuable insight that different results in tomato and eggplant plants might be affected by the density of eggs on differently-sized leaves of these two plant species. Maybe it would be possible to particularize this density differences in numbers?

Author Response

Reviewer 1

1) The novelty of the results remains doubtful. It is not the first study about the effects of Bemisia tabaci on either tomato or eggplant species. Therefore, it would be useful to review the already known data about B. tabaci effects on both plant species (tomato and eggplant) in Introduction. Especially, it should be stressed in which aspects this work is different from the previous works on the same topic.

Answer

We agree with the reviewer on the fact that several studies focus on the effects of Bemisia tabaci on tomato and eggplant; and, in fact, many of them are cited in our manuscript (and a few more have been added during the present revision of the text). However, it is important to underline that, in the light of the most recent knowledge, Bemisia tabaci is in fact a complex of genetically distinct species, having different biological characteristics, including the preference for host plants, the ability to transmit viruses, the acquisition of resistance to insecticides, etc. Most of the previous works on this topic have been realized on B. tabaci MEAM1, which is a different species than B. tabaci MED, the one we are focusing in the manuscript and that is presently increasing its frequency and abundance in Europe as a result of its biological characteristics (mainly its versatility in acquiring resistance to the most common insecticides used for whitefly chemical control). This aspect, alone, is an important novelty element of this manuscript, to which we can add the fact that the physiological and morphological parameters taken into consideration here, despite having some elements inevitably overlapped with previous works (however useful for appropriate comparisons of results), show others that are absolutely new and not taken into consideration in previous works.

This being considered, and following this comment from Reviewer 1, we added a few lines in the Introduction, stressing in which aspects this work is different from the previous ones on the same topic.

2) The arrangement of graphs in the Results, I think, should be changed. In the present version, for tomato, height is grouped with shoot dry weight (Fig. 2) and leaf area is grouped with chlorophyll content (Fig. 3); however, for eggplant, height is grouped with root dry weight and chlorophyll content (Fig. 4), and leaf area is grouped with shoot dry weight (Fig. 5). I suggest that uniformity between the results from two plant species would be helpful. Besides, I did not manage to find what happened to root dry weight in tomato.

Answer

The reviewer’s comment has been fully accepted and the graphs have been re-arranged accordingly.

3) At the end of Discussion, the authors make a valuable insight that different results in tomato and eggplant plants might be affected by the density of eggs on differently-sized leaves of these two plant species. Maybe it would be possible to particularize this density differences in numbers?

Answer

The reviewer’s comment has been fully accepted and additional data on the density of eggs on differently-sized leaves of both tomato and eggplant have been added in LL. 137-138 and LL. 213-216.

Reviewer 2 Report

The manuscript of Farina et al. describes the whitefly impact on morphology and physiology of tomato and eggplant under laboratory conditions. It was demonstrated that whiteflies of MED species are harmful to these crops, reducing several parameters that are essential to plant growth. Designs and analyses of experiments seemed appropriate, and the experiments appear to have conducted carefully. The presentation of methods and results were clear and well described. Based on my comments, the manuscript needs only minor revisions as outlined below.

L21-22: … the whitefly Bemisia tabaci (Gennadius) (Hemiptera: Aleyrodidae) is probably one of the main important pests that negatively …

L63: … of B. tabaci Mediterranean (MED) species …  

L189-190: … B. tabaci Middle East-Asia Minor 1 (MEAM1) species significantly …

Author Response

Reviewer 2

The manuscript of Farina et al. describes the whitefly impact on morphology and physiology of tomato and eggplant under laboratory conditions. It was demonstrated that whiteflies of MED species are harmful to these crops, reducing several parameters that are essential to plant growth. Designs and analyses of experiments seemed appropriate, and the experiments appear to have conducted carefully. The presentation of methods and results were clear and well described. Based on my comments, the manuscript needs only minor revisions as outlined below.

L21-22: … the whitefly Bemisia tabaci (Gennadius) (Hemiptera: Aleyrodidae) is probably one of the main important pests that negatively …

L63: … of B. tabaci Mediterranean (MED) species …

L189-190: … B. tabaci Middle East-Asia Minor 1 (MEAM1) species significantly …

Answer

All reviewer’s comments have been addressed and the text has been modified accordingly.

Reviewer 3 Report

The manuscript entitled: “Bemisia tabaci (Hemiptera: Aleyrodidae): what relationships  with and morpho-physiological effects on the plants it develops on?” submitted by Alessia Farina relates to the study of the morphological and physiological changes in plants of Solanum lycopersicum and Solanum melongena under Bemisia tabaci infestation.

Structure, language and style of the manuscript are generally acceptable. However, scientific sound of the work is low.  This is purely a descriptive study that measured only few variables in laboratory conditions. The study confirms existing views, commonly known facts and existing data. Research of a similar type was carried out already in the 90s of the last century. The scope of the research is narrow, experimental design (number of plants used) and results are very limited. Currently, such relationships are studied in more detail, based on the analysis of many parameters, and experiments are arranged in more replications.

Moreover, articles contained similar data have been published previously.

de Lima Toledo, C.A.; da Silva Ponce, F.; Oliveira, M.D.; Aires, E.S.; Seabra Júnior, S.; Lima, G.P.P.; de Oliveira, R.C. Change in the Physiological and Biochemical Aspects of Tomato Caused by Infestation by Cryptic Species of Bemisia tabaci MED and MEAM1. Insects 2021, 12, 1105. https://doi.org/10.3390/insects12121105

Islam, M. T.; Qiu, B.; Ren, S. Host preference and influence of the sweetpotato whitefly, Bemisia tabaci (Homoptera: Aleyrodidae) on eggplant (Solanum melongena L.). Acta Agriculturae Scandinavica Section B–Soil and Plant Science, 2010, 60(4), 320-325.

Schutze, I.X.; Yamamoto, P.T.; Malaquias, J.B.; Herritt, M.; Thompson, A.; Merten, P.; Naranjo, S.E. Correlation-Based Network Analysis of the Influence of Bemisia tabaci Feeding on Photosynthesis and Foliar Sugar and Starch Composition in Soybean. Insects 2022, 13, 56. https://doi.org/10.3390/insects13010056

In conclusion, this manuscript has not reached to the considerable scientific quality to be  published in Insects (a Q1 journal).

Author Response

Reviewer 3

The manuscript entitled: “Bemisia tabaci (Hemiptera: Aleyrodidae): what relationships with and morpho-physiological effects on the plants it develops on?” submitted by Alessia Farina relates to the study of the morphological and physiological changes in plants of Solanum lycopersicum and Solanum melongena under Bemisia tabaci infestation.

Structure, language and style of the manuscript are generally acceptable. However, scientific sound of the work is low. This is purely a descriptive study that measured only few variables in laboratory conditions. The study confirms existing views, commonly known facts and existing data. Research of a similar type was carried out already in the 90s of the last century. The scope of the research is narrow, experimental design (number of plants used) and results are very limited. Currently, such relationships are studied in more detail, based on the analysis of many parameters, and experiments are arranged in more replications.

Moreover, articles contained similar data have been published previously.

Answer

Basically, this comment focuses on two main elements:

  • Limited number of plants used in the experimental design;
  • Similarity of this work to previous studies.

As to the first aspect (Limited number of plants used in the experimental design), the examination of similar works allows to highlight a quite varied situation, which can be highlighted by the examples below:

Within such a scenario, we think that the methodology we adopted and the number of plants we used are fully around the average of what it is usually applied in similar works.

Considering the second aspect (Similarity of this work to previous studies), previous works deal mostly with other species of the B. tabaci complex and, in case they deal also with B. tabaci MED (as in de Lima Toledo et al., 2021, for example), they only consider physiological (e.g. gas exchange, chlorophyll fluorescence) and biochemical (enzymes, phenols and flavonoids only on leaves) aspects but not the combined effects of whitefly infestation on plant morphology. In particular, among previous works kindly suggested by the Reviewer 3 as example of similar works already published on the topic of our manuscript, Schutze et al. (2022) is totally dissimilar from our manuscript not only because it regards exclusively B. tabaci MEAM1 and considers only physiological (and not morphological) aspects, but especially because it has been realized on soybean (Fabaceae), which is a totally different plant than both tomato and eggplant (Solanaceae).

Round 2

Reviewer 3 Report

The corrections made by the authors to the manuscript and the response to my comments do not change my opinion about the work.

Author Response

According to the Rev 3 review report form, the manuscript was subjected to a further revision of the English language and style.